# CpG-Islands as Markers for Liquid Biopsies of Cancer Patients

**DOI:** 10.3390/cells9081820

**Published:** 2020-08-01

**Authors:** Maximilian Sprang, Claudia Paret, Joerg Faber

**Affiliations:** 1Department of Pediatric Hematology/Oncology, Center for Pediatric and Adolescent Medicine, University Medical Center of the Johannes Gutenberg-University Mainz, 55131 Mainz, Germany; paretc@uni-mainz.de (C.P.); joerg.faber@uni-mainz.de (J.F.); 2University Cancer Center (UCT), University Medical Center of the Johannes Gutenberg-University Mainz, 55131 Mainz, Germany; 3German Cancer Consortium (DKTK), site Frankfurt/Mainz, Germany, German Cancer Research Center (DKFZ), 69120 Heidelberg, Germany

**Keywords:** liquid biopsy, CpG islands, HCC

## Abstract

The analysis of tumours using biomarkers in blood is transforming cancer diagnosis and therapy. Cancers are characterised by evolving genetic alterations, making it difficult to develop reliable and broadly applicable DNA-based biomarkers for liquid biopsy. In contrast to the variability in gene mutations, the methylation pattern remains generally constant during carcinogenesis. Thus, methylation more than mutation analysis may be exploited to recognise tumour features in the blood of patients. In this work, we investigated the possibility of using global CpG (CpG means a CG motif in the context of methylation. The p represents the phosphate. This is used to distinguish CG sites meant for methylation from other CG motifs or from mentions of CG content) island methylation profiles as a basis for the prediction of cancer state of patients utilising liquid biopsy samples. We retrieved existing GEO methylation datasets on hepatocellular carcinoma (HCC) and cell-free DNA (cfDNA) from HCC patients and healthy donors, as well as healthy whole blood and purified peripheral blood mononuclear cell (PBMC) samples, and used a random forest classifier as a predictor. Additionally, we tested three different feature selection techniques in combination. When using cfDNA samples together with solid tumour samples and healthy blood samples of different origin, we could achieve an average accuracy of 0.98 in a 10-fold cross-validation. In this setting, all the feature selection methods we tested in this work showed promising results. We could also show that it is possible to use solid tumour samples and purified PBMCs as a training set and correctly predict a cfDNA sample as cancerous or healthy. In contrast to the complete set of samples, the feature selections led to varying results of the respective random forests. ANOVA feature selection worked well with this training set, and the selected features allowed the random forest to predict all cfDNA samples correctly. Feature selection based on mutual information could also lead to better than random results, but LASSO feature selection would not lead to a confident prediction. Our results show the relevance of CpG islands as tumour markers in blood.

## 1. Introduction

Classifiers based on global methylation profiles of cancer tissue were established in recent years and are in the works for several more entities. An example for global methylation classifiers is the classifier for central nervous system (CNS) tumours of the German Cancer Research Center (DKFZ). It works with Illumina methylation assays of fresh or formalin-fixed, paraffin-embedded (FFPE) material. It can distinguish the approximately 100 known tumour types of the CNS and was even shown to be more precise than conservative histological classification for some tumour types [1]. Another classifier for solid tumours was established by Wu et al., which can accurately diagnose and distinguish bone sarcomas: Ewing, chondro-, and osteosarcoma [2]. Both classifiers were based on random forests and used the CpG sites’ methylation b-values as features.

Liquid biopsies are likewise of emerging interest in the medical field in the last few years and have even started to be established in clinical use. The first liquid biopsy diagnostic tool was approved by the U.S. Food and Drug Administration (FDA) in 2016 and would identify small-cell lung cancer with mutations in the cell-free DNA (cfDNA) [3].

Liquid biopsies as a diagnostic or prognostic tool are a subject of research for many other tumour types and diseases. Not only genetic but also epigenetic biomarkers are currently under evaluation for cfDNA-based liquid biopsy. The methylation patterns of cfDNA have been shown to be useful for the identification and even localisation of different tumour types with a probabilistic method called CancerLocator, which uses self-computed CpG clusters [4].

Panagopoulou et al. were concerned with the methylation of certain loci typical for breast cancer but also took into account the level and size of cfDNA fragments. They were able to identify breast cancer patients with their liquid biopsy, utilising a logistic regression model as classifier [5].

In these works, CpG sites or clusters were used for classification. However, approximately 70% of annotated gene promoters are associated with CpG islands (CGI) which are, on average, 1000 base pairs (bp) long [6]. Importantly, CGIs acquire aberrant methylation in cancer, suggesting that cancer-specific CGI methylation can be distinguished from that in normal tissues [7].

This work aims to address the question of whether CpG islands rather than CpG clusters or sites can be used as a marker for cancers in liquid biopsies. As a model, we used hepatocellular carcinoma (HCC), since there were methylation data of cell-free DNA (cfDNA) samples from tumour patients available online. Between 75% and 85% of liver cancer incidences are HCC. There are around 600,000 deaths from HCC per year worldwide, the incidence as well as mortality being around two to three times higher in men than in women [8,9]. It is one of the leading causes of cancer-related deaths worldwide [10]. To this day, the orthotopic liver transplantation is the only curative procedure showing long-term curative effects. In early stages, the resection of tumour tissue can have curative effects, but it is seldom detected early enough [10]. Therefore, biomarkers for early detection are highly warranted. Here, we show that using as input methylation data from purified peripheral blood mononuclear cells (PBMCs) and HCC tumours, we could build a classifier that is able to distinguish the cfDNA of a HCC patient from healthy cfDNA.

## 2. Materials and Methods

### 2.1. Datasets

All datasets were publicly available and obtained from the Gene Expression Omnibus database. The authors providing the data to the Gene Expression Omnibus (GEO) database stated that informed consent was given by the patients as well as healthy individuals. The methylation and meta data cannot be used to identify or locate a tissue donor and should be fit for research purposes according to the declaration of Taipei.

Most of the sets were produced with an Illumina Infinium 450k bead chip, except for GSE130748 and GSE110185, which were produced using an Illumina Infinium EPIC bead chip. The cfDNA samples were extracted from plasma, ensuring that nearly only cell-free DNA is investigated. Both healthy cfDNA sample sets were taken from bigger sets in GEO and were used as controls in the corresponding experiments. The cancerous cfDNA was taken from an experiment that aimed to identify HCC liquid biopsies in the background of cirrhosis; we used the samples from patients with tumours (*n* = 22). No information about the tumour size and stage, fibrotic background, or Child–Pugh score were available. The MELD (Model of End Stage Liver Disease) score and AFP (Alpha-Fetoprotein) were available [11]. Importantly, the MELD score of the cfDNA samples was equally distributed between cirrhosis and HCC. Concerning AFP, 5 patients had a value more than 400 ng/mL, 4 patients had a value between 100 and 300 ng/mL and 11 were AFP-negative. PBMC DNA was extracted from a buffy coat, while the whole blood samples of both sets were not enriched before extraction. GSE77056 investigated the methylation profile of whole blood samples of crack cocaine dependents in comparison to non-addicts. We used the non-addict samples. GSE40279 contained 656 samples of whole blood methylation profiles from patients aged 19 to over 90. We used only the data of patients younger than 49; therefore, 101 samples remained. All together, 229 samples were used for our analysis: 10 cfDNA from healthy blood, 22 cfDNA from HCC, 35 HCC solid tumours, 125 whole blood samples, and 37 PBMCs. All samples are given in Table 1 with their respective GEO accession number.

### 2.2. Data Processing

The data stemming from different GEO datasets were imported and preprocessed in RnBeads, using its BMIQ normalisation implementation and array-combining capabilities to get combined island methylation data from all sets. RnBeads preprocessing steps include the filtering of CpG sites associated with single nucleotide polymorphisms (SNPs) and quality control of the datasets, which was done for each set independently. The normalisation was done with all samples in one set after combining them and discarding sites that do not occur in all samples. RnBeads computes the methylation value of a CpG island as a mean of all associated sites and names the islands after their location on hg19 [12,13].

The CpG island methylation data was processed in python. We used a random forest as classifying algorithm and tuned its hyperparameters with a random search. The random forest was tuned and trained with features previously selected by three different feature selection methods. In this work, we will also discuss the performance of these three feature selection techniques in combination with a random forest classifier for the application on methylation data. We used the scikit learn library [14].

### 2.3. Feature Selection

Depending on the training dataset, the feature selection method has a strong impact on the preciseness of prediction of the used classifier. We tested selection using the ANOVA F-value, mutual information selection, and selection from a logistic regression with l1-penalty (LASSO). The first are both filtering methods, while the LASSO selection is a so-called embedded feature selection method, which is a classification method itself and utilises the shrinkage of the penalty function to select the most important features. Filter methods take the dataset and compute values or thresholds for the features, by which the features are then selected. All selection methods were tested with a k of one to 10, with k being the number of features to select.

Both ANOVA and LASSO were used for methylation data in similar works. We decided to additionally test mutual information, since it is a measurement related to the gain methods the random forest uses to choose the optimal split.

### 2.4. Hyperparameter Tuning

The hyperparameters tuned included the number of estimators between 100 and 500, the max depth from 4 to 50 nodes, and if bootstrap aggregating should be used, or just conservative random selection. Additionally, the splitting criterion could be both information gain and Gini gain, and there were two splitting thresholds to choose: the minimum samples per leaf after a split between 2 and 100 and the minimum of samples per leaf before a split between 2 and 10. The area under curve of the receiver operating characteristic (AUROC) was used as a scoring function for optimising the random forest, and 500 iterations of randomised parameter settings in the given scales were done.

The scripts are available in a Github repository linked in the Appendix A.

## 3. Results

### 3.1. Principal Component Analysis Shows the Difference between Healthy and Cancerous cfDNA Samples

Figure 1 shows two scatterplots with the first two principal components (PCs) of a principal component analysis (PCA) as *x*- and *y*-axis. The datapoints display all 229 samples used in this work. The PCs are computed from CpG sites (a) as well as CpG islands (b) as features of the respective input data to the PCA. It can be used to get an overview of the data and see if the liquid biopsy datasets are distinguishable from the blood datasets with unsupervised techniques. It can be observed that the liquid biopsies appear closer to the blood than to the solid tumour samples, which is to be expected. Nevertheless, the liquid biopsy samples can be distinguished clearly from the PBMC as well as the whole blood data. However, the differences between the healthy (green) and cancerous (red) cfDNA are not as clear if the whole dataset is used as input.

It is also observable that the differences between the samples shrink if the CpG sites are computed to CpG island methylation: the difference between whole blood and PBMCs against the cfDNA from plasma samples is more distinct. Additionally, the scale of the scatterplot with the PCs computed from CpG sites is way larger. The greater the difference between two samples, the greater the difference in PC values.

This indicates that is easier to classify a sample using its CpG sites, rather than its islands; therefore, the feature extraction is of primary concern if a sample has to be classified using its island methylation.

### 3.2. Feature Selection and Supervised Analysis Can Differentiate between Liquid Biopsies of Healthy Individuals and HCC Patients

Since the PCA shows that the correct distinction of healthy and cancerous cfDNA samples will not be trivial, we employed three different feature selection techniques to shrink the dimension of the features in our dataset. High-dimensional data in relatively low sample sizes are hard to manage for machine learning applications. Therefore, data processing and feature selection are imperative for an efficient learning algorithm. In this work, we tested the ANOVA F-value, mutual information (MI), and LASSO feature selection methods. Given a set composed of all the datasets shown above, a random forest trained with features selected by all three feature selections was able to identify cfDNA samples as healthy or cancerous. The dataset was split randomly to a training set of 60% of the sample size and a test set of 40%. Using ANOVA in combination with random forest yielded an average accuracy score of 0.98 +/– 0.009 in a 10-fold cross validation, in which specificity was 0.96 +/– 0.048 and sensitivity was 0.97 +/– 0.038. Combined with MI selection, the classifier achieved an average accuracy of 0.98 +/– 0.017, showing a sensitivity of 0.99 and a specificity of 0.96 +/– 0.018 representing the best results in classifying the cancerous samples in this composition of data and a randomly split training and test set. With LASSO, the average accuracy in a 10-fold cross-validation was 0.98 +/– 0.018. Sensitivity was 0.98 +/– 0.019 and specificity was 0.98 +/– 0.023. The AUROC and average precision score are nearly 1 in all three combinations, which is due to using AUROC as a scoring function for the hyperparameter tuning. The average accuracy and F1 scores for all three selection methods in combination with a random forest are given in Figure 2a,b in comparison to the same predictions based on CpG sites instead of CpG islands. The F1 score is a combination metric of the positive predictive value, which is also called precision, and the sensitivity, which is often called recall.

Table 2 shows three reoccurring features that were selected by all three feature selections in multiple random states. The islands are given as their location on hg19. The corresponding genes are shown as well as the region relative to this gene. Additionally, the methylation status in the purified PBMCs is given as well as the methylation state of the solid HCCs. It is worth mentioning, that the healthy cfDNA had the same state as the PBMCs, while the cfDNA of HCC samples had a state that either was the same as that of the solid tumour (chr19:47614409-47614661) or resembled a mixture of both states (chr1:2979276-2980758). Non-methylated is referring to a beta value close to zero. Hypomethylated refers to beta values up to 0.25, semi-methylated refers to beta values from 0.25 to 0.75, and hypermethylated refers to beta values over 0.75. The highest beta values can reach is 1. It is important to keep in mind that the CpG islands methylation level is computed by computing the mean of all the CpG sites in the island. That means that the center could be thoroughly methylated, while the shore has a low methylation state, resulting in a semi-methylated value for the island. The methylated center could still have a regulatory impact, without the need to methylate the island thoroughly. Therefore, the methylation values of islands need to be assessed with nuance. cfDNA_6032742_Cirrhosis_with_HCC was the most mispredicted sample, which becomes clearer when looking at the PCA of the complete dataset (Figure 1), where it is located amidst healthy whole blood samples. Likewise, there are healthy cfDNA samples that reach wider into the cluster of cancerous cfDNA samples, which could account for the mispredictions of healthy cfDNA samples as cancerous ones. Other cfDNA samples of healthy and cancerous origin are mispredicted too in some runs, but this occurs significantly less frequently.

All information was taken from the UCSC (University of California in Santa Cruz) genome browser hg 19. The features were selected from the complete dataset. cfDNA of healthy individuals showed the same methylation state as PBMC samples. Likewise, the cancerous cfDNA showed the same state as the solid tumor samples, except for chr1:2979276-2980758, which shows a mixture of both states in different cfDNA samples.

### 3.3. A Random Forest Can Be Trained from PBMC Data and Solid Tumour Samples and Still Be Able to Predict cfDNA Samples Correctly

When using all samples save the liquid biopsy samples as a training set and testing on the liquid biopsy samples, only mutual information feature selection led to correctly predicted samples; it was still able to predict some of the samples correctly, but often mispredicted the majority of cfDNA samples of either healthy or cancerous class, depending on the chosen features. The other two feature selection methods could not provide a selection that would lead to acceptable results, predicting almost all cancerous cfDNA samples as healthy (data not shown). After removing the whole blood samples and just training on the solid tumours and PBMC samples, which are the main contributors of DNA fragments in the liquid biopsy samples that we wanted to predict, the ANOVA feature selection outperformed the other two. Mutual information still leads to a better result than random classifier with an AUROC over 0.5, but the mispredicted samples are always the same, with all different numbers of features selected. LASSO could not select CpG islands that enabled the classifier to correctly predict the cfDNA samples. However, ANOVA would always choose features that led to almost 100% accuracy with the cfDNA samples. At its lowest, three features selected led to 100% accurate results, meaning accuracy, sensitivity, and specificity values were all 1. These are the three methylation islands shown in Table 3. Figure 3 shows the prediction accuracy and F1 score of classifiers trained with features selected by the respective feature selections.

The results displayed were achieved with two PBMC samples removed (GSM3752974, GSM3752975). These were removed due to being strong outliers in the PCA. Interestingly, predictions from features selected by MI would increase their accuracy when these samples were not removed. The accuracy increased from the displayed 0.569 to 0.709 and the F1 score increased from 0.615 to 0.804. Combined with the other two selectors, the random forests performance increased when removing the two samples.

All information was taken from the UCSC genome browser hg 19. The features were selected from solid tumour and PBMC samples. cfDNA of healthy individuals showed the same methylation state as PBMC samples. Likewise, the cancerous cfDNA showed the same state as the solid tumor samples.

## 4. Discussion

Circa 2% of the genome contains high CpG density (approximately 1 CpG per 10 bp) in so-called CpG islands located at transcription start sites (TSS) within 50–60% of gene promoters. Typically, 5–10% of the CGIs are hypermethylated in cancer cells, leading to the permanent repression of tumour suppressor genes [6].

Several studies have reported the potential of investigating tumour-specific methylations in blood for the screening and diagnosis of cancer, mostly focusing on the analysis of specific promoters by quantitative Methylation-Specific PCR (qMSP); for example, the Epi proColon assay (Epigenomics, Inc.) that was approved by the FDA (US Food and Drug Administration) for screening colorectal cancer [15].

Here, we were able to build a classifier using methylation data from purified PBMCs and solid HCC tumours as input that could distinguish the cfDNA of a HCC patient from healthy cfDNA with an accuracy, sensitivity, and specificity of 1 using a random forest and features selected by ANOVA. The use of PBMC and HCC alone to build the classifier, without the need to include cfDNA profiles, is particularly interesting, since the abundance of cfDNA samples from plasma is still quite rare. To our knowledge, this is the first time that a classifier for cfDNA has been built by using merely PBMC and solid tumour data. Since this kind of data is free available to the scientific community for a large number of patients with different tumour entities via GEO and TCGA (The Cancer Genome Atlas Program), our approach can be easily tested in other cancer types. Other studies identified cancer-specific methylation sites by comparing the methylation profile of tumour and normal adjacent tissues [16], by including methylation data of plasma, but not PBMCs, in their algorithm [4], or by including methylation data of tumor cfDNA [11].

Our classifier worked by using the methylation of either CpG sites or island as input. It is difficult to make correct predictions based on methylation values when limited to using island methylation compared to the singular CpG sites, since information is lost when taking the mean of multiple CpG sites. This can already be observed using unsupervised methods such as PCA, where the difference in PC values shrinks significantly between the CpG sites as input and the CpG islands. However, singular CpG sites do not have a physiological impact that is yet known, while the islands in some cases can give insight on the physiology of the cancer from which the DNA originates.

The best performing features shown in Table 2 and Table 3 give examples for possible insights in tumour physiology gained from CGI methylation patterns. DEPDC5 has been suggest to act as a potential tumour suppressor in HCC, and the mRNA expression level of DEPDC5 is downregulated in HCC samples and correlates with poor patient prognosis [17]. While mutations and post transcriptional control have been suggested as mechanisms of DEPDC5 deficiency in a few cases [17], our results indicate that methylation could represent a general mechanism of the downregulation of DEPDC5 in HCC. The family of Aquaporines is also known to have links to tumourigenesis in different kinds of cancers. Interestingly, AQP6 was identified previously as significantly differentially methylated and downregulated in HCC compared with normal tissues [18]. Increasing the level of AZIN1 within the cells has a significant promoting effect on tumour growth and has been associated to kidney fibrosis [19]. High GCNT1 expression was shown to be associated with altered O-glycosylation in prostate cancer [20]. In all HCC datasets, the methylation state of this feature in solid cancer samples as well as in cancerous cfDNA was varying, but multiple samples showed low methylation, in turn increasing the expression of the gene [20]. ZC3H4 has been shown to promote epithelial-to-mesenchymal transition [21].

Thus, these features are interesting not only in their ability to lead a classifier to correctly predict the liquid biopsy samples, but they also could be used to identify functionally relevant players in HCC.

When being able to use cfDNA samples and solid tumours as well as healthy liquid biopsies, all feature selections in combination with a random forest would lead to solid predictions of all sample types, cfDNA included, with an accuracy between 0.96 and 0.99 as well as specificity from 0.96 to 0.98 and sensitivity between 0.98 and 1. Thus, if it is possible to obtain cfDNA samples of patients and healthy individuals alike, a prediction for diagnosis could be implemented. This has been already shown in the work of Hlady et al., showing that HCC can be diagnosed in a liquid biopsy even in the background of liver cirrhosis but using CpG sites as markers. That study contained and provided the cfDNA of HCC patients that were used in this work [11]. However, in that work, CpG sites and not islands were used. To our knowledge, we are the first to show that CpG island methylation status can be used as a marker in liquid biopsies. Additionally, regarding possible insights into the tumours’ physiology, using island methylation as a marker opens up possibilities for the cross-platform combination of methylation data, since classical methylation arrays do not have a CpG site resolution.

It should be mentioned that the mutual information feature selection is more versatile and less susceptible to changes in the source of its samples; in comparison to ANOVA and LASSO, it would predict with the highest accuracy if whole blood samples were inserted in the training phase, and its accuracy even increased when the two outlying PBMCs were not removed from the set. The ANOVA feature selection would lead to the features that are capable of correctly classifying the cfDNA as cancerous and healthy only with the solid tumour and PBMC dataset. Adding whole blood data or other PBMC sets (data not shown) would lead to a close to random predictor, illustrating the dependence of supervised learning methods on the training set. Mutual information would keep its accuracy or even increase it with a broader training set.

With both ANOVA and LASSO selections, the whole blood samples restricted a solid prediction of cfDNA samples when used as part of the training set. A possible explanation is that the whole blood samples contain relatively high amounts of non PBMC-stemming cfDNA, leading to the selection of features that are not sufficient for the prediction of cfDNA extracted from plasma.

A limitation of this work is the limited number of cfDNA HCC and healthy samples used (*n* = 22 and 10, respectively). Additionally, no pathological characteristics of the tumours were available. This information would be important to prove if our approach can identify only advanced tumours or also tumours at a very early stage. However, our algorithm was able to predict cfDNA samples correctly independently of the AFP value. Since the AFP value correlates with grade, progression, and survival [22], our algorithm seems to work independently from the pathological characteristics of the tumours. Importantly, in addition, cfDNA derived from AFP-negative HCC patients were correctly predicted. Finally, to allow the detection of other pathologies than cancer, the related methylation profile should be provided—for example, the methylation of cirrhosis samples—if the classifier has to distinguish between patients with cirrhosis, cancer, or healthy patients

## 5. Conclusions

Our data indicate that the development of biomarkers based on tumour-specific versus PBMC-specific island methylation is a promising approach that may be exploited to improve the early detection of cancer and disease monitoring of HCC and other tumours.

## Figures and Tables

**Figure 1 cells-09-01820-f001:**
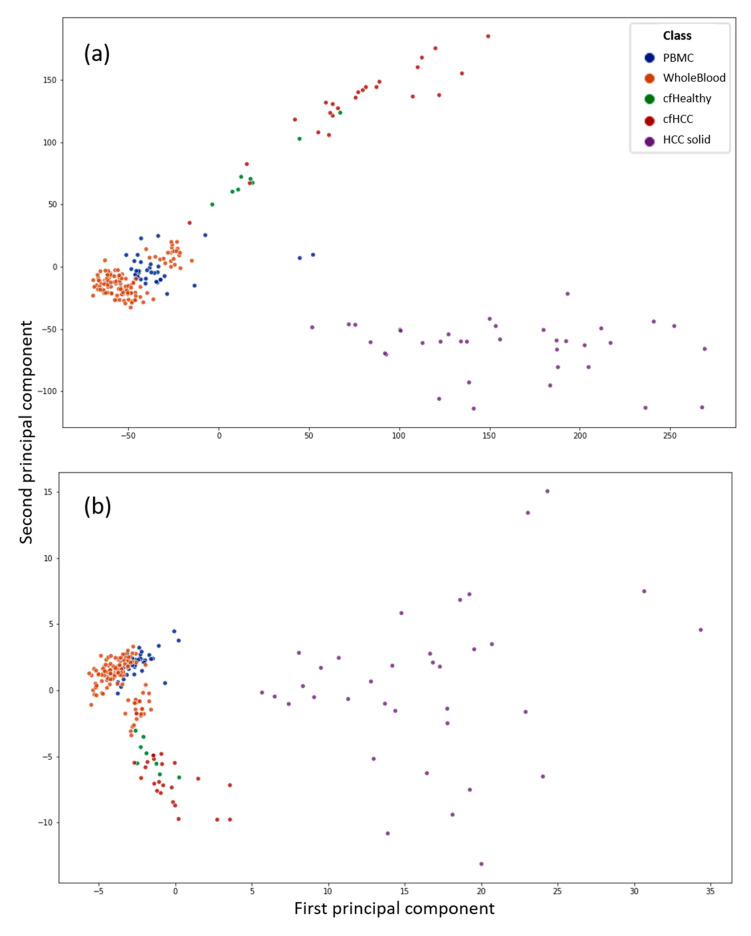
Principal component analysis (PCA) of the complete dataset. The features were in the form of CpG sites in (**a**) or computed CpG islands in (**b**). Blue is associated to purified peripheral blood mononuclear cell (PBMC) samples, while orange shows whole blood samples, where the complete DNA that can be found in a blood sample is extracted without further purification. The violet dots represent solid hepatocellular carcinoma (HCC) samples. Green and red are healthy and cancerous cfDNA samples extracted from plasma, respectively. The highest red dot in (**b**) located between the lower orange cluster is the sample cfDNA_6032742_Cirrhosis_with_HCC, which is the most mispredicted sample by all classifiers. In (**a**), it is the lowest red dot in proximity to the blood sample cluster.

**Figure 2 cells-09-01820-f002:**
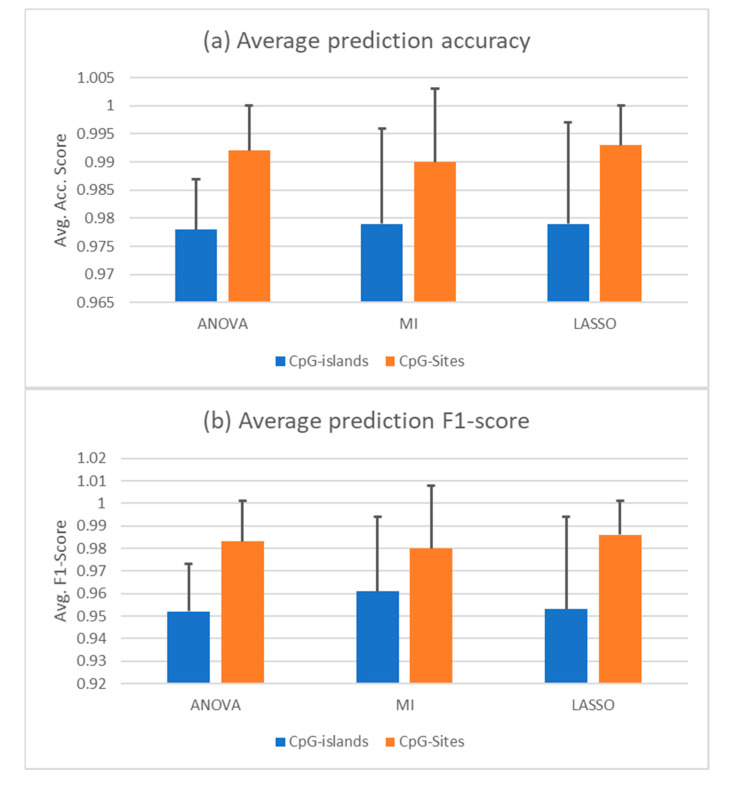
cfDNA classification based on random forest trained with 60% of complete data. (**a**) Average prediction accuracy of the random forests trained with—and tuned on—the features selected by the indicated methods. (**b**) Average F1 score of the same predictions. The *y*-axis indicated the score. Black bars are the standard deviation. The selectors were given the complete set of data as a randomly split training and test set with test sizes of 0.4. Ten iterations were conducted with different random states. CpG islands methylation (blue) or CpG sites (orange) were used for the analysis.

**Figure 3 cells-09-01820-f003:**
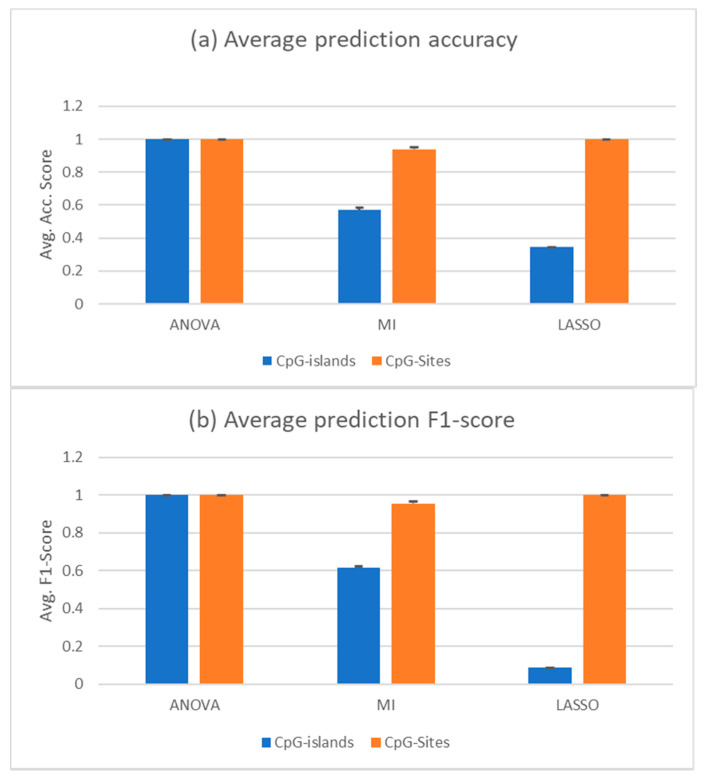
cfDNA classification based on random forest trained with all solid tumours and PBMC samples. The training set was limited to just the solid tumours and PBMC samples (35 and 37 samples, respectively). The selectors were trained only on solid tumour and PBMC samples as a randomly shuffled training set. The test set was composed of the healthy and cancerous cfDNA samples (10 and 22 samples, respectively). (**a**) Average prediction accuracy of the random forests trained with—and tuned on—the features selected by the indicated methods. (**b**) Average F1 score of the same predictions. Black bars are the standard deviation. Two PBMC samples were removed from the set, since they were outliers from the others in the PCA. CpG islands methylation (blue) or CpG sites (oranges) were used for the analysis.

**Table 1 cells-09-01820-t001:** Gene Expression Omnibus (GEO) datasets used in this work. cfDNA: cell-free DNA, PBMCs: peripheral blood mononuclear cells.

Sample	GEO Identifier	Exemplary Identifier in This Work	*n* =
Healthy blood: cfDNA	GSE110185	cf_Moss_x	8
Healthy blood: cfDNA	GSE122126	cfDNA_NCF_pool_x	2
Healthy blood: PBMCs	GSE130748	PBMC_x	37
Healthy blood: whole blood	GSE77056	blood_x	24
Hepatocellular carcinoma: solid tumour	GSE77269	HCC_2_x	20
Hepatocellular carcinoma: solid tumour	GSE99036	Hepatocellular carcinoma YSHxxx	15
Hepatocellular carcinoma: cfDNA	GSE129374	cfDNA_603xxxx_Cirrhosis_with_HCC	22
Healthy blood: whole blood	GSE40279	GEO Accession (GSM989xxx)	101

**Table 2 cells-09-01820-t002:** Frequently recurring features occurring in all feature selections and in multiple random states resulting in a high accuracy of prediction.

CpG Islands (Selected Features)	Genes	Region	Status in PBMC	Status in HCC
chr19:47614409-47614661	Zinc Finger CCCH-Type Containing 4, ZC3H4	Intron 2	Non-methylated	Semi-methylated
chr9:79073908-79074561	Beta-1,3-galactosyl-O-glycosylglycoprotein beta-1,6-Nacetylglucosaminyltransferase, GCNT1	Promoter Region, Exon 1, Intron 1	Non-methylated	Varying
chr1:2979276-2980758	PRDM16 Divergent Transcript		Hypermethylated	Semi-methylated

**Table 3 cells-09-01820-t003:** Minimum number of features selected by ANOVA leading to a high accuracy of prediction.

CpG-Islands (Selected Features)	Genes	Region	Status in PBMC	Status in HCC
chr12:50361368-50361652	Aquaporin 6, AQP6	Intron 1	Non methylated	hypomethylated
chr8:103875223-103877084	Antizyme inhibitor 1, AZIN1	Promoter, exon 1	hypomethylated	Non methylated
chr22:32149763-32150064	DEP domain-containing 5, DEPDC5	Promoter, exon 1	Non methylated	hypomethylated

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
