# Peer review of "CpG-Islands as Markers for Liquid Biopsies of Cancer Patients"

_cells, 2020, doi:10.3390/cells9081820_

Round 1

Reviewer 1 Report

Referee’s Comments for the Author:

The aim of this submitted manuscript was to evaluate the possibility of using global CpG-island methylation profiles as basis for prediction and prognosis by the help of liquid biopsies of cancer patients. Authors aimed to address the question if CpG-islands could be used as markers for cancers in liquid biopsies in model of hepatocellular carcinoma. Authors show the relevance of CpG-islands as tumor markers in blood with pontential high impact on the improvement of diagnosis and prognosis of cancer patients. A submitted manuscript is capable of being published after a minor revision process.

Major points:

Results; Section 3.2. Table 2: Authors could include information about similar methylation states in legnd in legend of this Tabel 2, particularly, declare that the healthy cfDNA had the same state as the PBMCs and the cfDNA of HCC samples had a state that was either the same like in the solid tumor.  

Minor points:

Abstract; Line 25: abbreviation PBMC should be explained as this is the first mention in the text

Methods; Line 93: There is no reference to  Table 1 in the text

Results; Line 192: s(t)ate

Table 2; correct alignment

Reviewer 2 Report

Liquid biopsies in cancer diagnosis and cancer screening are in urgent need, to screen fast and reliable larger population cohorts in general or patients at risk at least. 

Here the authors provided first evidences that tumour-specific methylation is or could be a promising approach, unfortunately, not more. CpG-islands methylation (blue) or CpG-sites (oranges) were used for the analysis. An interesting approach, surely.

However, after reading the manuscript still I could not figure out how good it is compared to other liquid biopsies in terms of sensitivity, specificity, negative prediction value, positive prediction value and so on. That's completely missing.

Unfortunately, this is from biggest interest to assess how good this novel liquid biopsy is actually performing compared to others. Additionally, I did not find any p value, any significancy given. The interested reader will need such performance parameters, that are common, making up his own thoughts about the value of this novel approach. No doubt, it sound interesting, of course, and we need new ideas in the field, therefore we need this values. And a power calculation. These standards we must keep. 

Additionally, I miss somehow the patients' demographics, maybe I overlooked it. Some information about tumour size and stage, fibrotic background, child-pugh-score etc..

What about other negative controls as other cancer entities, as intrahepatic CCA? Or non cancerous cirrhosis. What is the potential of this new approach, what are the limitations. What was the smallest HCC tumour that was detectable? Any correlations with tour size and load? Any gender specific differences or limitations. 

The discussion is to general. Afterwards I still did not know how the actual clinical performance was...some numbers should be included.  Some PIs have the tendency just to read the abstract and discussion. Keep it in mind.

Overall, a major revision is needed to allow me to assess the manuscript fully and to understand its performance compared to other current experimental liquid biopsies methodologies. I do not want to discourage the authors, since we must think out of the box, but every new idea has to be comparable to others.

I miss any information about ethical approval that was granted...which authority, approval number etc...its a must... patient consent etc.. Helsinki declaration... were ethical principles kept?

Round 2

Reviewer 2 Report

I got the point that this present study is somehow a proof of concept and I assume it will likely yield into a further grant application to dig deeper into it.

As said, the KI must been able to differentiate between HCC with or without cirrhosis vs cirrhosis only and other cancer entities as iCCC, otherwise its just a nice gimmick. Time will proof.

Overall, I agree on acceptance in order to push your research that I fully support.

Well done,

Dr. Miroslaw Kornek